# Serum Biomarker Signatures of Choroid Plexus Volume Changes in Multiple Sclerosis

**DOI:** 10.3390/biom14070824

**Published:** 2024-07-10

**Authors:** Dejan Jakimovski, Robert Zivadinov, Ferhan Qureshi, Murali Ramanathan, Bianca Weinstock-Guttman, Eleonora Tavazzi, Michael G. Dwyer, Niels Bergsland

**Affiliations:** 1Buffalo Neuroimaging Analysis Center, Department of Neurology, Jacobs School of Medicine and Biomedical Sciences, University at Buffalo, State University of New York, Buffalo, NY 14203, USA; 2Wynn Hospital, Mohawk Valley Health System (MVHS), Utica, NY 13502, USA; 3Center for Biomedical Imaging at the Clinical Translational Science Institute, University at Buffalo, State University of New York, Buffalo, NY 14203, USA; 4Octave Bioscience, Menlo Park, CA 94025, USA; 5Department of Pharmaceutical Sciences, Jacobs School of Medicine and Biomedical Sciences, University at Buffalo, State University of New York, Buffalo, NY 14214, USA; 6Department of Neurology, Jacobs Comprehensive MS Treatment and Research Center, Jacobs School of Medicine and Biomedical Sciences, University at Buffalo, State University of New York, Buffalo, NY 14203, USA; 7Multiple Sclerosis Centre, IRCCS Mondino Foundation, 27100 Pavia, Italy

**Keywords:** choroid plexus, serum biomarkers, multiple sclerosis, GFAP, NfL

## Abstract

Increased choroid plexus (CP) volume has been recently implicated as a potential predictor of worse multiple sclerosis (MS) outcomes. The biomarker signature of CP changes in MS are currently unknown. To determine the blood-based biomarker characteristics of the cross-sectional and longitudinal MRI-based CP changes in a heterogeneous group of people with MS (pwMS), a total of 202 pwMS (148 pwRRMS and 54 pwPMS) underwent MRI examination at baseline and at a 5-year follow-up. The CP was automatically segmented and subsequently refined manually in order to obtain a normalized CP volume. Serum samples were collected at both timepoints, and the concentration of 21 protein measures relevant to MS pathophysiology were determined using the Olink™ platform. Age-, sex-, and BMI-adjusted linear regression models explored the cross-sectional and longitudinal relationships between MRI CP outcomes and blood-based biomarkers. At baseline, there were no significant proteomic predictors of CP volume, while at follow-up, greater CP volume was significantly associated with higher neurofilament light chain levels, NfL (standardized β = 0.373, *p* = 0.001), and lower osteopontin levels (standardized β = −0.23, *p* = 0.02). Higher baseline GFAP and lower FLRT2 levels were associated with future 5-year CP % volume expansion (standardized β = 0.277, *p* = 0.004 and standardized β = −0.226, *p* = 0.014, respectively). The CP volume in pwMS is associated with inflammatory blood-based biomarkers of neuronal injury (neurofilament light chain; NfL) and glial activation such as GFAP, osteopontin, and FLRT2. The expansion of the CP may play a central role in chronic and compartmentalized inflammation and may be driven by glial changes.

## 1. Introduction

Multiple sclerosis (MS) is a chronic neuroinflammatory, demyelinating, and neurodegenerative disease of the central nervous system (CNS) that results in substantial and long-term accumulation of physical and cognitive disabilities [1]. In addition to the well-described peripheral immune/lymphocyte activation, the propagation and maintenance of the inflammatory process within the CNS are less known. Accumulating evidence suggests that the later stages of the disease are characterized by a compartmentalized inflammation that is driven, at least in part, by microglial constituents [2].

Recent findings have energized the pursuit toward better understanding the role of the choroid plexus (CP) in MS pathophysiology [3]. As a highly vascularized structure, the CP may represent one of the main gateways through which the inflammatory cells transverse and establish continuity between the peripheral and cerebral immune changes [4,5]. Moreover, the CP’s role in maintaining cerebrospinal fluid (CSF) homeostasis could further contribute toward MS-specific changes [6]. The lack of CSF absorption and filtration could additionally prolong the exposure of the periventricular brain to proinflammatory cytokines and toxic compounds such as ceramides [7]. MRI-derived measures of CP volume and its microstructural properties have recently been linked with both clinical and radiological outcomes in MS [3]. For example, people with MS (pwMS) have a significantly larger CP than matched healthy controls; a greater CP volume and CP expansion are associated with increased chronic lesion activity and greater brain atrophy [8]. Even after adjusting for all established demographic, clinical, and MRI-based predictors of worse MS outcomes, the inflammation within the CP is independently associated with (contributing toward) a greater rate of clinical disability progression [3].

The development of the latest fourth-generation, ultra-sensitive, ELISA-based assays has enabled reliable quantification of picomolar concentrations in various bodily fluids [9]. Although readily measurable in the CSF, the serial monitoring for diagnostic and prognostic purposes is both invasive and generally infeasible in clinical settings. Through the CP and other brain–blood interfaces, a very small fraction of the total concentration of brain-specific and pathology-based molecules diffuses into the blood stream. Damage to various CNS cells results in the release of cell-specific molecules such as neurofilament light chain (NfL; biomarker of axonal/neuronal injury) and glial fibrillary acidic protein (GFAP; biomarker of glial injury and activation) [10]. Using these sensitive assays, the quantification of these molecules in the serum has enabled researchers and clinicians to develop proteomic prognostic algorithms that correlate with the current and future extent of MS pathology [11]. In particular, NfL and GFAP blood-based biomarkers have been previously associated with acute inflammatory and chronic neurodegenerative changes in MS that are directly related to worse long-term outcomes. For example, pwMS with higher NfL or GFAP levels exhibit increased physical/cognitive disability, greater incidence of gadolinium-enhancing lesions, and greater whole-brain atrophy [12,13].

Based on this background, we aimed at understanding the blood-based biomarker characteristics of the MRI-based CP volume changes in a heterogeneous MS population. We hypothesized that changes within the CP are associated with biomarkers of glial activation and have characteristic signature when compared to the well-known pathophysiology.

## 2. Materials and Methods

### 2.1. Study Population

All study participants were part of a larger 5-year longitudinal study that investigated the cardiovascular, environmental, and genetic factors in MS (CEG-MS) [14]. Previous biomarker analysis on this particular population was published [15]. The inclusion criteria for this particular 5-year analysis were (1) being diagnosed as MS per the 2017 McDonald criteria [16], (2) having blood-drawn samples available, and (3) having an MRI scan available within close proximity to the blood draw (within 30 days). The exclusion criteria were (1) having an acute clinical worsening of the disease (relapse) 30 days before the study investigations, (2) having received intravenous corticosteroid therapy 30 days before the study investigations, and (3) being pregnant or a nursing mother. Since acute infections can result in an increase in symptomatology and confound serum analysis, pwMS with any acute health changes were also excluded from the study.

All participants underwent comprehensive clinical examination at both study timepoints by an experienced and NeuroStatus-certified investigator. The disability level of the pwMS was determined using the Expanded Disability Status Scale (EDSS) score [17]. All demographic and clinical outcomes were collected using a standardized questionnaire and additionally referenced using the electronic medical records (EMRs). Based on the clinical presentation and the history of the disease, the pwMS were classified as either people with relapsing-remitting MS (pwRRMS) or people with progressive MS (pwPMS) based on the Lublin criteria [18]. The disease-modifying therapy (DMT) used by the pwMS was categorized based on its anti-inflammatory potency. This study was approved by the Institutional Review Board (IRB) of the University at Buffalo, and all participants provided a written consent form at both study timepoints.

### 2.2. MRI Acquisition and Analysis

All participants underwent MRI scanning at both timepoints using the same 3T GE Signa Excite HD 12.0 scanner (GE, Milwaukee, WI, USA). There were no major hardware or software changes between the two scans for any of the participants. For the purpose of the CP analysis, two sequences were utilized: (1) two-dimensional (2D) T2 fluid-attenuated inversion recovery (FLAIR) sequence with an echo time (TE)/inversion time (TI)/repetition time (TR) of 120/2100/8500 ms, echo train length (ETL) of 24, and flip angle of 90 degrees; and (2) three-dimensional (3D) T1-weighted imaging (WI) that utilized fast, spoiled, gradient echo (SPGR) with a magnetization-prepared inversion recovery pulse with parameters of TE/TI/TR = 2.8 ms/900 ms/5.9 ms and flip angle of 10°. A full description of the MRI protocol is published elsewhere [3].

The segmentation of the CP at both timepoints was performed using the recon-all stream from FreeSurfer software, version 6 [19]. Subsequently, the initial segmentations were refined using a Gaussian mixture model method [20] and then manually reviewed and corrected as necessary. Normalized CP volume was derived by multiplication with the SIENAX-based scaling factor to control for head size. All analyses were performed in a fully blinded manner with respect to clinical and demographic characteristics.

### 2.3. Blood-Based Biomarker Analysis

The blood-based biomarker analysis was performed in partnership with Octave Bioscience (Menlo Park, CA, USA). Serum samples from both timepoints were analyzed in a single batch and those performing the analyses were blinded to any demographic or clinical characteristic. The samples were originally preprocessed within 24 h of the blood draw and stored at −80 °C at the School of Pharmacy and Pharmaceutical Sciences at the University at Buffalo. A custom-made Multiple Sclerosis Disease Activity (MSDA) assay based on the Olink™ platform and Proximity Extension Assay methodology was used. An in-depth validation of the assay was published elsewhere [11]. In particular, the assay quantifies the concentration of 21 proteins that have been implicated in the key pathophysiological MS pathways (immunomodulation, neuroinflammation, myelin biology, and neuroaxonal integrity). The list of proteins, their commonly used aliases, and their participation in the aforementioned pathways are shown elsewhere [15].

### 2.4. Statistical Analyses

All statistical analyses were performed using SPSS version 28.0 (IBM, Armonk, NY, USA). The data distribution was determined using visual inspection of the histograms and the Q-Q plots. Normally distributed data are presented using mean and standard deviations (SDs), whereas non-normally distributed data are presented using medians and interquartile ranges (IQRs). For the purpose of satisfying the linear regression assumptions of normal distribution, all biomarker data were logarithmically transformed. Categorical variables were compared using the chi-square test, normally distributed numeric variables using Student’s *t*-test, and non-normally distributed numeric variables using the Mann–Whitney U test. Data with Poisson distribution were compared using negative binomial regression. Adjusted comparisons were performed using analysis of covariance (ANCOVA). The relationship between baseline CP measures and their change over the follow-up were assessed using Pearson’s correlations.

The statistical analysis utilized a two-block linear regression model that explored the relationship between the CP volume and the biomarkers (as independent predictors). In particular, the first block force included three demographic characteristics of age, sex, and BMI in the model regardless of whether they were significant predictors of the CP measure. A second step-wise block was added in which all blood-based biomarkers were added as predictors, but only the significant ones that survived were added into the final model. The change in R2 is reported for each step-wise predictor added. The tolerance and collinearity of the predictors were determined through the variance inflation factor (VIF), where VIF ≥ 2.5 was considered multicollinear. Tolerance for multicollinearity ranged from 0 to 1, with a tolerance below 0.4 considered as a potential risk of multicollinearity. *p*-values lower than 0.05 were considered as statistically significant.

## 3. Results

### 3.1. Demographic and Clinical Characteristics of the Study Population

The demographic and clinical characteristics of the study population are outlined in Table 1. A total of 202 pwMS were enrolled in the study (148 pwRRMS and 54 pwPMS). As expected, the pwPMS were significantly older (55.3 vs. 44.1 years old, *p* < 0.001), had longer disease duration (21.7 vs. 11.1 years, *p* < 0.001), and were more disabled at both study visits (*p* < 0.001 for both). The pwRRMS had significantly greater numbers of relapses over the follow-up when than the pwPMS (0.204 vs. 0.09, *p* < 0.001). There were no differences in the DMT distribution at both visits between the MS phenotypes. In terms of CP measures, there were no significant differences between the pwRRMS and pwPMS groups at both baseline and follow-up timepoints. The pwPMS had numerically greater CP volume when compared to the pwRRMS. The absolute blood-based biomarker values and differences between the MS phenotypes are shown in Appendix A Table A1 and published elsewhere [14].

### 3.2. Cross-Sectional and Longitudinal Relationships between Choroid Plexus and Blood-Based Biomarkers

The cross-sectional associations between baseline CP measures and baseline proteins are shown in Table 2. After adjusting for age, sex, and BMI, there were no significant blood-based predictors of the CP volume at baseline (model not created). At the follow-up visit, higher CP volume was significantly associated with higher serum NfL levels (standardized β = 0.373, *p* = 0.001) and lower osteopontin levels (standardized β = −0.23, *p* = 0.02). The protein measures increased the explained variance in the change in CP volume from 9.5% to 19% (R2 change *p* < 0.001). There were no multicollinearity concerns between the predictors (VIF < 1.63). There were no changes in an additional model that added the DMT as a covariate, with the same predictors remaining significant (DMT was not a significant predictor of CP volume).

The longitudinal associations of the baseline blood-based measures and the change in CP volume over the follow-up are also shown in Table 2. After adjusting for age, sex, and BMI, a greater % CP volume expansion was associated with higher baseline GFAP levels (standardized β = 0.277, *p* = 0.004) and lower baseline FLRT2 levels (standardized β = −0.226, *p* = 0.014). The blood-based biomarkers increased the explained variance in the change in CP volume from 0.8% to 8.1% (R2 change *p* < 0.001). When performed in pwRRMS and pwPMS separately, the baseline serum FLRT2 levels remained a significant predictor of % CP volume change only in the pwPMS group (standardized β = −0.462, *p* = 0.01). There were no significant associations between concurrent change in CP volume and change in proteomic measures over the follow-up period.

## 4. Discussion

This study provides detailed blood-based biomarker characterization of the CP volumetric changes, a measure that is increasingly being noted as a potential predictor of MS outcomes. In particular, the association between cross-sectional CP volume and higher NfL levels may indicate pre-existing pathology that has resulted in an increase in both of these damage-related measures. On the other hand, the predictive value of GFAP and FLRT2 toward future increases in CP volume may suggest that the glial changes are inductive factors for this MS process. The corroborating factors from the literature are further discussed hereafter.

When compared to our MS population, an analysis of 108 cognitively unimpaired volunteers from the Baltimore Longitudinal Study of Aging (average age of 55.7 years old) demonstrated that a higher CP volume was cross-sectionally significantly associated with GFAP levels (r = 0.22, *p* = 0.002) but not with serum NfL levels (*p* = 0.35) [21]. This study further investigated the microstructural integrity of the CP and demonstrated that it was significantly associated with pTau181, NfL, and GFAP levels, suggesting an age-related decline in the CP structure that may be related to neurodegenerative and glial etiology [21]. In general, we believe that both blood-based measures are intrinsically linked with the CP changes in both aging and MS pathology, with the timeline discrepancies being only a result of different sample sizes (study power) and methodological variations within the literature. Our findings also further corroborate the published positron emission tomography (PET) results that demonstrated greater translocator protein (TSPO) 18F-DPA-714 uptake (indicative of microglial activity) in the CP of pwMS when compared to healthy controls [22,23]. CP volume was additionally correlated with greater PET uptake [22]. Additional histopathological analysis also confirmed a large population of CD163+ phagocytes in the CP that could colocalize the greater PET uptake [23]. A proteomic atlas of the developing and maturing CP has been recently published. While embryonic CP structures contain glial precursors and immature neurons, the adult CP contains GFAP-expressing astrocytes that form the CP stromal space [24]. No MS-specific proteomic classification of the CP has been published as of yet that would allow the assessment of the disease-induced deviations in CP proteomics.

The heterogeneous associations of osteopontin levels with different MS outcomes can be explained by the pleotropic biological nature of MS. In this analysis, lower osteopontin levels were related to greater CP volume and may be explained by osteopontin’s ability to polarize B cells toward a less activated phenotype [25]. In vitro stimulation of activated B cells with osteopontin resulted in suppression of co-stimulatory signals and downregulation of pro-inflammatory cytokine secretion (IL-6) [25]. The lack of osteopontin may induce upregulation that may result in higher CP volume. Contrarily, osteopontin is also a potential aggregator of B cells in MS tissue, with an ability to form lymphocytic tertiary clusters that have been characterized in pwMS [25,26]. This dual nature of osteopontin may produce an early negative association with CP volume but may later induce an increase in CP changes (initial suppression of the B-cell activity leading to the later promotion of cell aggregation). While we did not identify osteopontin as a predictor of CP volume change over the follow-up, the exact role of osteopontin and its relationship to CP changes and/or B-cell aggregation still remain fairly unknown. Over the past 10 years, osteopontin has been intermittently suggested as an important biomarker in MS, with associations with greater gray matter pathology, relapses, and progressive phenotype [14,27]. The relevance of FLRT2 serum concentrations in CP changes is further corroborated and validated by the fact the same biomarker was singled out as a potential predictor of clinical MS disability and radiological worsening [15]. Serum FLRT2 levels were among the few isolated predictors of global and regional brain atrophy in MS [15]. Lastly, brain injury can result in the upregulation of reactive astrocytes that express higher FLRT2 levels, and it is correlated with higher expression of GFAP [28].

Highly effective DMTs can potentially suppress CP expansion when compared to other anti-inflammatory DMTs with lower efficacy or when compared to that in pwMS that were not treated with any DMT [29]. In our sample, the effectiveness classification of the DMTs did not influence the size of the CP at baseline or influence the 5-year change. However, an indication bias may be present, where active MS cases would necessitate high-potency DMT, whereas aging pwMS commonly discontinue their DMT due to a lack of inflammatory activity [30].

This study has several limitations that should be emphasized. First, this study did not include healthy controls that would have provided sufficient anchoring of the study findings. Although our study was longitudinal in nature, the blood-based protein levels and CP changes may occur at different timescales. For example, the changes in proteins at baseline may be short-lived and fluctuate over time, whereas the CP changes may occur slowly over a much longer period of time. Serial investigations, on a shorter interval, will be crucial in determining the true relationship between changes in blood-based biomarkers and the changes in CP volume.

## 5. Conclusions

In conclusion, the increase in CP volume is associated with neuronal injury markers such as NfL and may be predicted by inflammatory blood-based markers of glial activation such as GFAP, osteopontin, and FLRT2. These findings further suggest that CP may play a central role in the chronic and compartmentalized inflammation that is driven by glial changes.

## Figures and Tables

**Table 1 biomolecules-14-00824-t001:** Demographic and clinical characteristics of the study population.

Demographic and Clinical Characteristics	pwMS (n = 202)	pwRRMS (n = 148)	pwPMS (n = 54)	*p*-Value
Female, n (%)	151 (74.8)	106 (71.6)	45 (83.3)	0.09 ^a^
Age at baseline, mean (SD)	47.1 (11.1)	44.1 (10.6)	55.3 (7.9)	**<0.001 ^b^**
Time of follow-up, mean (SD)	5.4 (0.6)	5.4 (0.6)	5.5 (0.6)	0.732 ^b^
BMI at baseline, mean (SD)	27.5 (5.8)	27.9 (6.2)	26.5 (4.5)	0.1 ^b^
Age of disease onset, mean (SD)	32.9 (9.8)	32.6 (9.0)	33.6 (11.8)	0.6 ^b^
Disease duration at baseline, mean (SD)	13.4 (10.2)	11.1 (8.5)	21.7 (10.5)	**<0.001 ^b^**
EDSS at baseline, median (IQR)	2.5 (1.5–5.0)	1.5 (1.5–2.5)	6.0 (4.0–6.5)	**<0.001 ^c^**
EDSS at follow-up, median (IQR)	3.0 (1.6–6.0)	2.0 (1.5–3.5)	6.5 (4.0–6.5)	**<0.001 ^c^**
EDSS absolute change, mean (SD)	0.4 (0.9)	0.4 (0.9)	0.4 (0.7)	**<0.001 ^b^**
Disability progression, n (%) *	56 (30.9)	38 (28.6)	18 (37.5)	0.251 ^a^
Relapse rate over the follow-up, mean (SD)	0.172 (0.369	0.204 (0.4)	0.09 (0.24)	**<0.001 ^d^**
DMT at baseline, n (%)				
IFN-β	85 (42.1)	60 (40.5)	25 (46.3)	0.271 ^a^
Glatiramer acetate	37 (18.3)	24 (16.2)	13 (24.1)	
Natalizumab	29 (14.4)	25 (16.9)	4 (7.4)	
Off-label DMT	5 (2.5)	3 (2.0)	2 (3.7)	
No DMT	46 (22.8)	36 (24.3)	10 (18.5)	
DMT at follow-up, n (%)				
IFN-β	68 (33.7)	52 (35.1)	16 (29.6)	0.797 ^a^
Glatiramer acetate	45 (22.3)	31 (20.9)	14 (25.9)	
Natalizumab	15 (7.4)	12 (8.1)	3 (5.6)	
Oral DMT	28 (13.9)	22 (14.9)	6 (11.1)	
Off-label DMT	12 (5.9)	8 (5.4)	4 (7.4)	
No DMT	34 (16.8)	23 (15.5)	11 (20.4)	
CP volume at baseline, mean (SD)	2.59 (0.9)	2.54 (0.9)	2.7 (0.88)	0.862 ^e^
CP volume at follow-up, mean (SD)	2.68 (0.8)	2.6 (0.7)	2.9 (1.1)	0.406 ^e^
CP volume % change, mean (SD)	9.2 (31.9)	9.0 (33.5)	10.0 (27.8)	0.487 ^e^

**Legend:** pwMS—people with multiple sclerosis, pwRRMS—people with relapsing remitting multiple sclerosis, pwPMS—people with progressive multiple sclerosis, BMI—body mass index, EDSS—Expanded Disability Status Scale, DMT—disease-modifying therapy, IFN—interferon, SD—standard deviation, IQR—interquartile range. ^a^—Chi-square test, ^b^—Student’s *t*-test, ^c^—Mann–Whitney U test, ^d^—negative binomial regression, ^e^—age-adjusted analysis of covariance (ANCOVA). *p*-value lower than 0.05 was considered statistically significant and shown in bold. *—available for 173 pwMS.

**Table 2 biomolecules-14-00824-t002:** Baseline blood-based predictors of cross-sectional choroid plexus volume and changes over the follow-up.

Predictors of CP Volume at Follow-Up	R2	Standardized B	t-Statistics	*p*-Value	95% CI LB	95% CI UP	Tolerance	VIF
Age	0.095	−0.066	−0.751	0.454	−434.5	195.8	0.987	1.013
Sex		0.087	0.859	0.392	−8.9	22.5	0.758	1.319
BMI		−0.077	−0.829	0.409	−36.9	15.2	0.905	1.105
Log10 NfL	0.146	0.373	3.324	**0.001**	548.4	2169.9	0.613	1.630
Log10 OPN	0.19	−0.230	−2.366	**0.020**	−2250.8	−198.5	0.814	1.228
**Predictors of CP % Volume Change**	**R2**	**Standardized B**	**t-Statistics**	***p*-Value**	**95% CI LB**	**95% CI UP**	**Tolerance**	**VIF**
Age	0.008	−0.111	−1.286	0.201	−0.853	0.181	0.899	1.112
Sex		0.021	0.249	0.804	−10.21	13.151	0.957	1.045
BMI		0.115	1.283	0.202	−0.35	1.641	0.831	1.203
Log10 GFAP	0.04	0.277	2.914	**0.004**	12.4	65.0	0.737	1.357
Log10 FLRT2	0.081	−0.226	−2.492	**0.014**	−105.8	−12.2	0.812	1.232

**Legend:** CP—choroid plexus, BMI—body mass index, FLRT2—fibronectin leucine-rich repeat transmembrane protein 2, OPN—osteopontin, GFAP—glial fibrillary acidic protein, CI—confidence intervals, LB—lower bound, UP—upper bound, VIF—variance inflation factor. *p*-value lower than 0.05 was considered statistically significant and shown in bold.

## Data Availability

The data utilized in this manuscript are available upon reasonable request from the corresponding author.

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
