# Peer review of "Serum Biomarker Signatures of Choroid Plexus Volume Changes in Multiple Sclerosis"

_biomolecules, 2024, doi:10.3390/biom14070824_

Round 1
Reviewer 1 Report
Comments and Suggestions for Authors
This is a study that examines the changes in the volume of the choroid plexus and the correlates those changes with several proteomic biomarkers, including NfL, GFAP, osteopontin, etc. The background for this study is that expansion of the choroid, a very vascular structure, correlates with disease progression and possibly with inflammatory markers. The main criticism of this reviewer is that it would be helpful to examine how NfL changes correlate with other MRI markers such as gadolinium-enhancing lesions and volume change. How does volume change is the choroid plexus change with these other markers associated with markers such as NfL.
Minor points: In the discussion, do the authors really mean ptau191, or should this be ptau181, a well-described marker for Alzheimer disease neurodegeneration.
The final sentence of the discussion is a sentence fragment and needs to be completed.
Overall, this paper does lend credence to the hypothesis that choroid plexus volume can help predict disease course in MS, but the authors likely have data which could reinforce this point.
Author Response
Reviewer 1:
This is a study that examines the changes in the volume of the choroid plexus and the correlates those changes with several proteomic biomarkers, including NfL, GFAP, osteopontin, etc. The background for this study is that expansion of the choroid, a very vascular structure, correlates with disease progression and possibly with inflammatory markers. The main criticism of this reviewer is that it would be helpful to examine how NfL changes correlate with other MRI markers such as gadolinium-enhancing lesions and volume change. How does volume change is the choroid plexus change with these other markers associated with markers such as NfL.
Response: We thank the Reviewer for the great comment. We have expanded the Introduction to include the vast literature that demonstrates associations of NfL with MRI markers such as gad-enhancing lesions and brain volume change.
Minor points: In the discussion, do the authors really mean ptau191, or should this be ptau181, a well-described marker for Alzheimer disease neurodegeneration.
Response: We thank the Reviewer for noticing the typo. This is now corrected.
The final sentence of the discussion is a sentence fragment and needs to be completed.
Response: The sentence is now corrected in the manuscript.
Overall, this paper does lend credence to the hypothesis that choroid plexus volume can help predict disease course in MS, but the authors likely have data which could reinforce this point.
Response: We have further re-emphasized the relationship between choroid plexus volume and changes in MS outcomes. This information was previously published elsewhere:
Bergsland N, Dwyer MG, Jakimovski D, Tavazzi E, Benedict RHB, Weinstock-Guttman B, et al. Association of Choroid Plexus Inflammation on MRI With Clinical Disability Progression Over 5 Years in Patients With Multiple Sclerosis. Neurology. 2023;100(9):e911-e20.
Reviewer 2:
The research is generally well thought out and focuses on a relevant number of patients. However, in the inclusion criteria, it would be advisable for the authors to check whether the patients had been on treatment for one year or more (to ensure that the disease was controlled with the treatment that each patient had). In the exclusion criteria, they should check whether the patients were in the absence of acute processes (such as colds, gastroenteritis, etc.) that could interfere with laboratory determinations.
Response: The pwMS enrolled in this study did not have any acute ongoing process. Typically, acute changes such as non-MS related infection are accompanied with worsening of MS symptoms and therefore not included in our study. This was clarified in the manuscript.
The design and statistical study are correct.
However, there is an important misconception or nomenclature error. The title refers to proteomics. From there to the conclusion section, proteomics is mentioned. However, no proteomic techniques have been used and no proteomic approach has been taken. No two-dimensional electrophoresis, mass spectrometry or specific bioinformatics techniques have been used (for example). The determinations made (Nfl) or GFAP protein do not require a proteomics laboratory. GFAP protein (and many other biomarkers) can be measured by ELISA and Nfl can be measured by SIMOA (modified digital ELISA) in any laboratory with such equipment.
Therefore, although molecules of protein lineage have been measured, one should not speak of proteomics because proteomic techniques have not been used. Consequently, the title and all allusions to the field of proteomics should be changed. It would be more correct to refer to serum biomarkers just by name.
Response: We have corrected the use of the term “proteomic” within the entire manuscript. Moreover, we have changed the title of the manuscript to reflect these changes.
Reviewer 2 Report
Comments and Suggestions for Authors
The research is generally well thought out and focuses on a relevant number of patients. However, in the inclusion criteria, it would be advisable for the authors to check whether the patients had been on treatment for one year or more (to ensure that the disease was controlled with the treatment that each patient had). In the exclusion criteria, they should check whether the patients were in the absence of acute processes (such as colds, gastroenteritis, etc.) that could interfere with laboratory determinations.
The design and statistical study are correct.
However, there is an important misconception or nomenclature error. The title refers to proteomics. From there to the conclusion section, proteomics is mentioned. However, no proteomic techniques have been used and no proteomic approach has been taken. No two-dimensional electrophoresis, mass spectrometry or specific bioinformatics techniques have been used (for example). The determinations made (Nfl) or GFAP protein do not require a proteomics laboratory. GFAP protein (and many other biomarkers) can be measured by ELISA and Nfl can be measured by SIMOA (modified digital ELISA) in any laboratory with such equipment.
Therefore, although molecules of protein lineage have been measured, one should not speak of proteomics because proteomic techniques have not been used. Consequently, the title and all allusions to the field of proteomics should be changed. It would be more correct to refer to serum biomarkers just by name.
Author Response

(The authors gave the same response as above.)
